# Stress during the COVID-19 Pandemic Moderates Pain Perception and Momentary Oxytocin Levels

**DOI:** 10.3390/jcm12062333

**Published:** 2023-03-16

**Authors:** Ekaterina Schneider, Dora Hopf, Monika Eckstein, Dirk Scheele, Corina Aguilar-Raab, Sabine C. Herpertz, Valery Grinevich, Beate Ditzen

**Affiliations:** 1Center for Psychosocial Medicine, Institute of Medical Psychology, Heidelberg University Hospital, Bergheimer Str. 20, 69115 Heidelberg, Germany; 2Institute of Psychology, Faculty of Behavioral and Cultural Studies, Heidelberg University, 69117 Heidelberg, Germany; 3Department of Social Neuroscience, Faculty of Psychology, Ruhr-University Bochum, 44791 Bochum, Germany; 4Center for Psychosocial Medicine, Department of General Psychiatry, Heidelberg University Hospital, 69115 Heidelberg, Germany; 5Department of Neuropeptide Research in Psychiatry, Central Institute of Mental Health, Medical Faculty Mannheim, Heidelberg University, 68159 Mannheim, Germany

**Keywords:** stress, physical pain, emotional pain, oxytocin, ecological momentary assessment, COVID-19

## Abstract

Self-reported pain levels have been associated with increased stress levels during the COVID-19 pandemic. Less is known about the long-term effects of stress on individuals’ physical and emotional pain levels and their associations with the neuropeptide hormone oxytocin. We aimed to predict momentary pain through individual stress levels and momentary oxytocin levels at genuinely high-stress phases, namely during COVID-related lockdowns. In a cross-sectional (*n* = 254) and a longitudinal (*n* = 196) assessment during lockdowns in Germany, participants completed a 2-day ecological momentary assessment (EMA) protocol (collecting six saliva samples on two consecutive days each and simultaneously reporting on stress, physical, and emotional pain levels) in 2020, as well as one year later, in 2021. Hierarchical linear modeling revealed significant positive associations between individuals’ stress levels and physical pain, both cross-sectionally (b = 0.017; t(103) = 3.345; *p* = 0.001) and longitudinally (b = 0.009; t(110) = 2.025; *p* = 0.045). Similarly, subjective stress ratings showed significant positive associations with emotional pain on a within-person (b = 0.014; t(63) = 3.594; *p* < 0.001) as well as on a between-person (b = 0.026; t(122) = 5.191; *p* < 0.001) level. Participants further displayed significantly lower salivary oxytocin when experiencing higher levels of emotional pain (b = −0.120; t(163) = −2.493; *p* = 0.014). In addition, high-stress levels significantly moderated the association between physical pain and salivary oxytocin (b = −0.012; t(32) = −2.150; *p* = 0.039). Based on mechanistic and experimental research, oxytocinergic mechanisms have long been suggested to modulate pain experiences, however, this has not yet been investigated in everyday life. Our data, which was collected from a large sample experiencing continued stress, in this case, during the COVID-19 pandemic, suggests that individuals experience more intense physical pain and elevated stress levels, as shown by particularly low salivary oxytocin concentrations.

## 1. Introduction

The outbreak of the COVID-19 pandemic made it necessary to implement immediate restrictions and physical distancing to control the spread of the virus. The threat of the virus in combination with restrictions and quarantines has been associated with increased rates of anxiety, depression, stress, and decreased well-being [1,2,3,4]. In parallel, self-reported pain increased [5,6,7], especially when patients suffering from chronic pain did not have access to treatment [8], suggesting that adequate clinical care can prevent pain augmentation during a pandemic [9,10]. Further, self-reported pain during the pandemic was associated with decreased physical activity and increases in psychological stress [6].

Initial research has demonstrated that stress per se modulates pain perception and vice versa, that is, the association between stress and pain seems to be bidirectional [11,12]. However, while acute stress can reduce pain perception, chronic stress, in contrast, has been reported to increase pain (for review see [13,14]; for reports on sex differences see [15,16]). Moreover, negative emotions, as well as their intensity, seem to modulate this relationship [17]. Research regarding emotional pain, however, is very scarce [18], despite the knowledge that the assessment and understanding of emotional pain can have important implications in clinical work [19]. One previous study found that previously experienced emotional pain (e.g., childhood abuse, parent’s psychopathology) has detrimental effects on medical pain conditions and increases depression and anxiety later in life [20].

Data on neuroendocrine correlates of pain suggest a critical role of oxytocin, as the exogenously administrated neuropeptide elicits analgesic effects and increases pain tolerance [21]. More specifically, in men with chronic back pain, oxytocin administration attenuated pain intensity when acute thermal pain was applied, but not when back pain was occurring spontaneously [22], indicating that the analgesic effects of oxytocin might depend on the type of pain. Likewise, in a study using heat pulses generated by an infrared laser that selectively activates Ad- and C-fiber nerve endings [23], analgesic effects of oxytocin have been found, but not for noxious thermode heat stimuli [24]. Interestingly, the anti-nociceptive effects of oxytocin appear to be sex-specific and have been mostly observed for men (e.g., [25], but also see [26]).

The literature on the associations between peripheral oxytocin and pain is scarce. For example, elevated oxytocin concentration was reported in migraineurs [27], whereas, in children with abdominal pain and inflammatory bowel disease [28] and patients with low back pain [29], oxytocin levels were decreased. In line with this, another study suggests that, in women, higher oxytocin was linked to higher pain tolerance [30]. Interestingly, recent studies demonstrated that pain, as well as oxytocin, can be modulated simultaneously by social factors. Storytelling [31] or the mother’s voice [32] can reduce pain and increase oxytocin concentrations in children, possibly due to the neuropeptide’s stress-reducing effects. However, to the best of our knowledge, there is currently no study investigating momentary and long-term effects of stress on pain and their associations with oxytocin concentrations in daily life during the pandemic. Furthermore, studies on emotional pain are underrepresented, in general, and even less is known about the role of emotional pain in everyday life and its associations with stress and salivary oxytocin. To fill this gap, we conducted an ecological momentary assessment (EMA) in everyday life during the first lockdown in Germany and repeated the assessment one year later during the second lockdown to probe long-term effects. We specifically investigate distinct associations with physical pain, as compared to emotional pain. Before this study, we hypothesized that self-reported stress would be associated with increased individual pain levels on a momentary level within one assessment phase (1) and one year later (2). Furthermore, we expected a negative association between oxytocin and pain on a momentary level (3), and that this link would be moderated by individuals’ subjective stress levels (4).

## 2. Materials and Methods

### 2.1. Study Design and Population

During the first lockdown in Germany, which began in March 2020, we launched the initial part of our study to investigate the impact of physical contact restrictions on individuals’ well-being (data published elsewhere [33,34]). During the first lockdown between April and August 2020, a total sample of 247 participants completed a 2-day EMA (t1); 254 participants (*n* = 196, plus *n* = 58 newly recruited) completed a second 2-day EMA (t2) in 2021. 

Participants were recruited via social media, local newspapers, and radio and were included for participation if they were over 18 years of age and German-speaking. In addition to participating in an online questionnaire assessment, interested individuals were invited to take part in a 2-day psychobiological ecological momentary assessment (EMA). At six time points per day, participants collected saliva samples via the passive drool method and simultaneously assessed their momentary subjective data. Sampling time points were adapted based on individual wake-up times and were taken directly after awakening, 30 min after, 45 min after, 2 ½ hours after, 8 h after, and directly before going to sleep. All participants received the informed consent documents as well as the collecting devices via mail. Participants received standardized instructions on the saliva collection devices, the use of their smartphone to collect momentary ratings and proper storage of the samples. The experimenters provided individual instructions via e-mail and by phone, constantly monitored the daily assessments and were available to be contacted for questions. One year after the completion of the first EMA study (t1), participants were asked to participate in the second EMA study (t2) with an identical study design. Of the 472 individuals who originally expressed interest in the study, 257 were enrolled. Ten individuals did not finish data collection due to personal reasons (e.g., psychological burden, loss of a family member, etc.) resulting in 247 individuals with a complete dataset. In 2021, of the 288 interested individuals, 259 were enrolled in the study. Further, 254 individuals completed the two-day data collection, whereas 5 individuals withdrew for personal reasons. 

The ethics committee of Heidelberg University Medical Faculty approved the study (approval no. S-214/2020) and all participants provided written informed consent. The study was registered online athttps://drks.de/search/en/trial/DRKS00021671 (accessed on 5 December 2022).

### 2.2. Measurements

#### 2.2.1. Subjective Ratings of Ecological Momentary Assessment in t1 and t2

In both years, t1 and t2, participants received links via short messages on their phones to collect subjective ratings. Momentary levels of stress were assessed through single items (“Please indicate how you feel at the moment…”) using visual analog scales from 0 (relaxed) to 100 (stressed). Pain levels were assessed in the second year of EMA (t2) only, with the questions “Do you feel any physical or emotional pain at this moment?” and “If yes, please indicate the intensity of your experienced physical/emotional pain” ranging from 1 (little pain) to 6 (extreme pain). Additionally, participants were asked where in the body they felt the pain. Momentary stress levels were assessed at four of six time points: 45 min, 2 ½ hours, 8 h after waking up; as well as directly before going to bed, whereas pain levels were assessed at the first and last measurement time points per day (directly after awakening and directly before going to bed) resulting in 1016 possible pain measures (2 times per day, from 254 individuals in 2 days) and 3600 stress measures (4 times per day, from 196 and 254 individuals on 2 days in 2020 and 2 days in 2021, respectively).

#### 2.2.2. Neuroendocrine Measures

After receiving the corresponding link, participants self-sampled their saliva using Salicaps^®^ (IBL International, Hamburg, Germany) (small plastic tubes). Immediately after the collection of each sample, participants stored them in their home freezers until the study team collected the samples on dry ice. The saliva samples were stored at −80 °C until the biochemical analyses were performed at the Institute of Medical Psychology’s biochemical lab at Heidelberg University Hospital.

For the analyses of endogenous oxytocin concentrations, saliva samples were thawed and centrifuged at 4 °C at 1500× *g* for 15 min and subsequently analyzed following the protocol of oxytocin enzyme-linked immunosorbent assay from Enzo Life Sciences (ELISA; ENZO Life Sciences, Lörrach, Germany) with a detection limit of 15 pg/mL. The analyses were performed without extraction and variation coefficients for intra- and inter-assay precision were 6.12% and 11.13%, respectively, for the analyses of the first year (t1), whereas in the second year (t2) the coefficients were 5.9% and 13.63%, respectively.

### 2.3. Statistical Analyses

For data processing and statistical analyses, we used IBM SPSS version 27 and R studio (R version 4.1.1). We conducted multiple hierarchical linear models to test whether momentary stress levels were associated with the intensity of experienced physical and emotional pain in everyday life. To separate within-person and between-person effects, self-reported stress was centered around each person’s mean and the person’s mean was centered on the grand mean for both years (t1 and t2, respectively). 

First, we included momentary stress levels during the second lockdown (t2) controlling for age, sex, being in a relationship (yes/no), and measurement time points as independent variables to predict momentary self-reported pain levels during the second lockdown (t2). Subsequently, we examined whether higher stress levels during the first lockdown would predict the intensity of experienced pain one year later. Therefore, in this model, we included the grand mean centered stress ratings of the first lockdown (t1) instead of the stress ratings of the second lockdown (t2). 

Next, we tested the hypothesis that momentary oxytocin would be associated with the intensity of experienced emotional and physical pain. To normalize the distribution, oxytocin levels were log-transformed (natural logarithm). We included the person mean, as well as the grand mean centered intensity of experienced pain, as predictors and oxytocin levels as the dependent variable. For models including oxytocin measures as dependent variables, we additionally controlled for body mass index (BMI), momentary food and drink intake, alcohol, caffeine, and cigarette consumption, medication intake, as well as physical activity and teeth brushing. 

Finally, we tested whether individual stress levels moderated the link between momentary pain and oxytocin. Oxytocin levels were included as the dependent variable, whereas stress levels (t2), the intensity of pain, the product term (stress (t2) x intensity of pain), and the aforementioned control variables were included as predictors in the model. 

For all these models, we also conducted random intercept and random slope models and compared the fit of these models to models without random slope using Likelihood Ratio tests. Since none of the random slope models showed a better model fit (*p* > 0.414), we report only the random intercept models. 

## 3. Results

To test our first, third, and fourth hypotheses we used the whole t2 sample (t2all), whereas for the analysis of the second hypothesis we only used the subgroup of individuals, who participated in our study twice. The mean age of the sample was 34.07 (SD = 13.06) ranging between 19 and 79 years (for more details of sample characteristics please see Table 1). Of the whole sample 9.6% indicated that they have been diagnosed with depression, 3.1% with multiple diagnoses, and 1.9% with posttraumatic stress disorder. 

Of the whole sample, 154 participants (60.6%) with a mean age of 33.34 (SD = 12.99) reported physical pain, whereas 147 participants (57.9%) with a mean age of 34.08 (SD = 13.03) reported emotional pain at any time point of the 2-day EMA. The most frequent forms of physical pain were 1) head, neck, or shoulder at 31.3%, followed by a combination of different types of pain at 30.7% and back pain at 12.9%. The overall mean for the intensity of physical pain was 1.64 (SD = 0.76) and 1.85 (SD = 0.92) for emotional pain indicating that most individuals experienced low levels of pain. Mean stress levels were M = 37.46 and showed a broad range (SD = 18.89).

### 3.1. Self-Reported Stress Levels as Predictors for Physical and Emotional Pain Intensity

Results from separate random intercept multilevel analyses showed that on a between-person level, self-reported stress (t2) was significantly and positively associated with the intensity of physical pain (b = 0.017; t(103) = 3.345; *p* = 0.001), but not on a within-person level (b = 0.004; t(48) = 1.007; *p* = 0.319). Similarly, individual stress levels during the first lockdown in 2020 (t1) predicted subsequent physical pain levels during the second lockdown in 2021 (t2) on a between-person level (b = 0.009; t(110) = 2.025; *p* = 0.045) (for more details please see Table 2A). 

For emotional pain, we found a significant positive association with subjective stress ratings on within-person (b = 0.014; t(63) = 3.594; *p* < 0.001), as well as between-person (b = 0.026; t(122) = 5.191; *p* < 0.001) levels. Individual stress levels during the first year of the lockdown (t1), however, showed only marginal association with emotional pain in the subsequent year (t2) (b = 0.009; t(101) = 1.858; *p* = 0.066) (for more details please see Table 2B).

### 3.2. Associations of Oxytocin Levels with the Intensity of Pain, and Its Interaction with Self-Reported Stress (t2)

The analysis of our third hypothesis revealed a significant negative main effect of emotional pain and oxytocin concentrations (b = −0.120; t(163) = −2.493; *p* = 0.014), as well as a negative statistical trend association of physical pain intensity and oxytocin concentrations on a within-person level (b = −0.105; t(141) = −1.867; *p* = 0.064), but not on a between-person level. In addition, to analyze whether stress levels moderated the association between pain intensity and oxytocin concentrations, we included self-reported stress levels (t2), as well as the interaction variable of pain x stress (t2) as additional predictors. We found a significant association between momentary stress levels (b = −0.010; t(32) = −3.254; *p* = 0.003) and a significant interaction of physical pain x stress (b = −0.012; t(32) = −2.150; *p* = 0.039), with oxytocin concentrations showing a negative association of physical pain and oxytocin in individuals with high levels of stress (see Figure 1). No significant associations on a between-person level were found. In none of the analyses did we detect any effect of sex.

## 4. Discussion

In the present study, we investigated short and long-term associations between self-reported stress levels and the intensity of physical and emotional pain in the general population during the COVID-19 pandemic. Additionally, we analyzed whether salivary oxytocin levels were associated with the intensity of pain and whether this association was moderated by stress levels. 

Results of our longitudinal psychobiological EMA study during the COVID-19 lockdowns revealed that individuals with higher stress levels are more likely to report higher intensity of physical and emotional pain in daily life, and that higher stress levels during the first lockdown predicted higher pain intensity during the second lockdown. These findings are in line with recent studies reporting that increased psychological stress was associated with pain augmentation in times of pandemic [6] and elevated levels of stress predicted higher rates of low back pain one year later [35]. On a within-person level, we found that higher momentary stress levels predicted emotional pain levels. However, in contrast to our hypothesis and previous findings, which showed that psychosocial stress induction can increase physical pain sensitivity [36,37], we did not find a within-person association between subjective stress and physical pain levels. 

As for psychobiological mechanisms underlying the association of stress and pain, we found a significant association between momentary emotional pain levels and oxytocin concentrations, indicating that oxytocin concentrations were lower when emotional pain was higher. With regard to physical pain, there was a marginally significant negative association between the momentary intensity of physical pain and salivary oxytocin. Interestingly, analyses showed that momentary self-reported stress levels significantly interacted with the intensity of pain to predict oxytocin levels. Thus, individuals experiencing more intense physical pain and elevated stress levels show particularly low salivary oxytocin concentrations during the COVID-19 pandemic. These results support previous findings reporting a negative association between oxytocin, pain and stress in human subjects [28,29]. For example, Anderberg and Uvnäs-Moberg found that patients suffering from fibromyalgia syndrome had lower plasma oxytocin concentrations when their stress and pain ratings were high [38]. In line with previous studies [26], we did not find any role of sex in the association of oxytocin and pain. Although in animal models, oxytocin has been shown to directly decrease pain signaling to the spinal cord [39,40], it has also been suggested to elicit stress-induced hyperalgesia [13]. Furthermore, our observation of stress as a moderator of endogenous oxytocin effects is consistent with a previous study showing that higher levels of plasma oxytocin were associated with fewer depressive symptoms and more sensitive maternal behavior among women who reported high levels of psychosocial stress [41]. As such, stress seems to moderate oxytocin effects in various domains. 

Our study has several strengths and limitations that need to be addressed. To minimize the drop-out rate during the study, we assessed individual stress and physical pain using single items, which led to a relatively large sample size in an ecologically valid setting. However, these items might not comprehensively reflect the individual’s experience of stress and pain during times of pandemic, as standard questionnaires would. Moreover, these data might be influenced by self-report bias, since standard or experimental induction of stress or pain was not used in this study. To investigate the long-term effects of the pandemic on the psychobiological state, we conducted a 1-year follow-up assessment (t2) of our initial assessment (t1). Thus, we were able to analyze whether initial stress levels during the first lockdown predicted pain intensity in the following year. Unfortunately, we did not assess pain ratings during the first lockdown, which makes it impossible for us to study the bidirectional link between stress and pain in the long-term and to analyze whether high pain levels in the first year would also predict higher self-reported stress in the following year. On the one hand, the restrictions and lockdowns, resulting from the pandemic, affected most of the population creating a unique social situation to investigate the relationship between stress, pain, and oxytocin in a relatively controlled setting. Thus, including the general population with a broad variance of momentary stress in our study made the interpretation of the results more generalizable. On the other hand, the relatively low pain ratings in our sample make it difficult to extrapolate our findings to patients with high pain intensity. To the best of our knowledge, this is the first study investigating emotional pain and its associations with subjective stress as well as oxytocin levels. Emotional pain might overlap with general psychological burden, and thus its association with stress does not seem surprising, but the data collected might contribute to a more fine-grained analysis of neuroendocrine factors in emotional and physical pain. As for the limitations of the mode of collection of the neuroendocrine factor, it is possible that peripheral oxytocin measured in saliva or blood may not necessarily reflect oxytocin release in the brain [42]. However, in stressful situations such positive correlations have been reported in animal studies [43], thus supporting the notion that peripheral oxytocin may signal corresponding responses with central stress and pain-related mechanisms. 

## 5. Conclusions

Taken together, our findings indicate that individuals experiencing more stress might be at higher risk of experiencing more pain in times of pandemic. Moreover, individual stress levels were predictive for future pain ratings showing that individuals with higher stress during the initial phase of the pandemic reported more intensive pain one year later. Additionally, we found that momentary oxytocin levels were decreased in individuals experiencing more stress and higher pain intensity. This suggests that subjectively experienced stress during a pandemic can impair the analgesic effects of oxytocin. Thus, reducing stress and promoting healthy coping strategies in stressful situations might reduce pain perception.

## Figures and Tables

**Figure 1 jcm-12-02333-f001:**
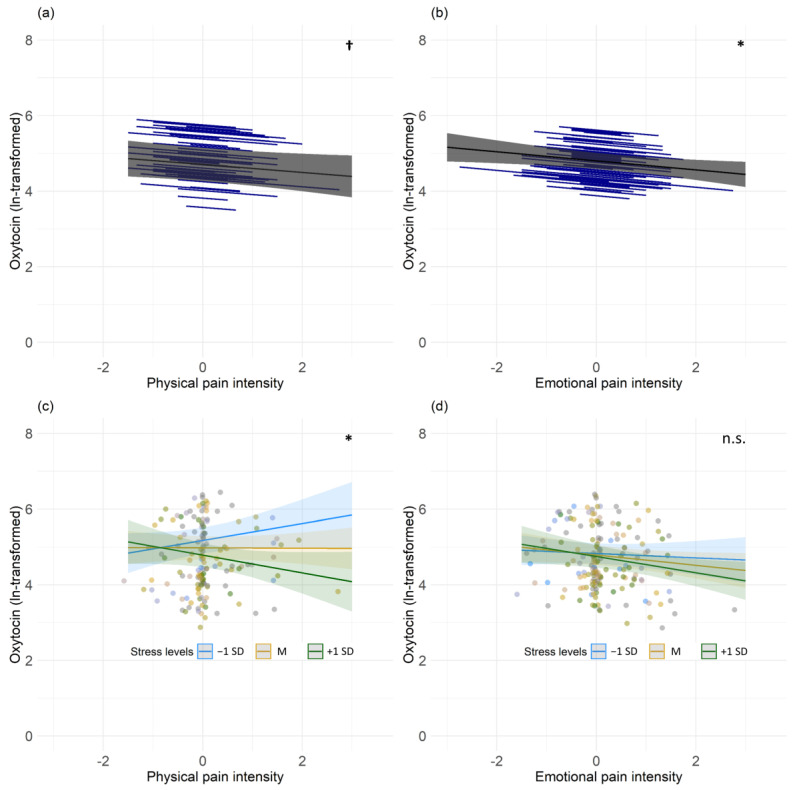
Associations of oxytocin levels with pain intensity and its interaction with stress. Panel (**a**) illustrates the associations of person’s mean-centered physical pain and (**b**) emotional pain intensity with oxytocin concentrations (pg/mL) on a within-person level. The grey lines indicate the overall predicted slope, whereas the blue lines indicate the individual’s predicted slopes with their minimum and maximum predicted values as endpoints. The grey area depicts 95% confidence band. Panels (**c**) and (**d**) illustrate individual stress levels as a moderator of the associations between pain intensities and oxytocin (pg/mL) concentrations. Colored lines represent the effect of stress on the association between pain and oxytocin using mean (M) and 1 standard deviation below (−1 SD) and above (+1 SD). Each dot represents individual values. * represents significant results, whereas † represents a statistical trend and n.s. a non-significant moderation.

**Table 1 jcm-12-02333-t001:** General characteristics of the sample.

Ecological Momentary Assessment in 2021						
	t2 Men (*n* = 76)		t2 Women (*n* = 176)		t2 all (*n* = 254 *)	
	M	SD	M	SD	M	SD
Age (years)	33.89	13.91	34.23	12.67	34.07	13.06
Oxytocin (pg/mL) ^a^	133.66	113.61	135.27	117.31	135.437	116.95
Stress levels ^b^	32.84	18.04	39.26	19.13	37.46	18.93
Physical pain intensity ^b^	1.42	0.49	1.73	0.82	1.64	0.76
Emotional pain intensity ^b^	1.71	0.66	1.88	1	1.85	0.92
**Repeated Measures of Ecological Momentary Assessment**						
	**t1 (*n* = 196)**		**t2 (*n* = 196)**			
	**M**	**SD**	**M**	**SD**		
Age (years)	32.72	12.94	33.86	13.05		
Oxytocin (pg/mL) ^a^	160.02	117.56	130.05	114.24		
Stress levels ^b^	33.00	17.13	36.78	19.46		
Physical pain intensity ^b^			1.66	0.79		
Emotional pain intensity ^b^			1.87	0.92		

Note. Table depicts means (M) and standard deviations (SD). Number of participants indicated as (*n*). ^a^ Oxytocin raw values, before ln-transformation and ^b^ momentary self-reports (using a visual analogue scale from 0 (relaxed) to 100 (stressed) and a Likert scale ranging from 1 (little pain) to 6 (extreme pain). * Including individuals (*n* = 2) without indication of their sex.

**Table 2 jcm-12-02333-t002:** Hierarchical linear models of the associations between stress, pain, and oxytocin.

**(A) Physical Pain Intensity**				
**Effects**	**Stress(t2) → Physical Pain (t2)**	**Stress(t1) → Physical Pain (t2)**	**Physical Pain (t2) → Oxytocin (t2)**	**Physical Pain (t2) × Stress(t2) → Oxytocin (t2)**
**Fixed Effects**				
Intercept	**2.057 (0.427); *p* < 0.001**	**1.207 (0.310); *p* < 0.001**	**4.986 (0.374); *p* < 0.001**	**4.777 (0.788); *p* < 0.001**
Stress (t2) ^a^	0.004 (0.004); *p* = 0.319	**---**	**---**	**−0.010 (0.003); *p* = 0.003**
Physical pain ^b^	**---**	**---**	−0.105 (0.056); *p* = 0.064	−0.049 (0.090); *p* = 0.593
Stress (t2) x Physical pain (t2)	**---**	**---**	**---**	**−0.011 (0.004); *p* = 0.040**
**Between-person**				
Stress (t1) ^a^	**---**	**0.009 (0.004); *p* = 0.045**	**---**	**---**
Stress (t2) ^a^	**0.017 (0.005); *p* = 0.001**	**---**	**---**	−0.002 (0.004); *p* = 0.703
Physical pain (t2) ^b^	**---**	**---**	−0.056 (0.087); *p* = 0.524	−0.030 (0.122); *p* = 0.804
Stress (t2) x Physical pain (t2)	**---**	**---**	**---**	−0.000 (0.004); *p* = 0.982
**Covariates**				
Age (t2)	0.009 (0.007); *p* = 0.242	**0.013 (0.006); *p* = 0.043**	−0.008 (0.005); *p* = 0.124	**−0.019 (0.007); *p* = 0.004**
Sex ^c^	0.190 (0.231); *p* = 0.413	0.334 (0.182); *p* = 0.069	0.067 (0.153); *p* = 0.662	0.052 (0.201); *p* = 0.798
Partner (t2) ^d^	−0.315 (0.239); *p* = 0.191	−0.268 (0.191); *p* = 0.164	0.143 (0.165); *p* = 0.387	**0.481 (0.217); *p* = 0.029**
Time point (t2) ^e^	**−0.053 (0.021); *p* = 0.014**	−0.002 (0.014); *p* = 0.895	−0.013 (0.012); *p* = 0.267	−0.003 (0.017); *p* = 0.837
Body Mass Index (t2)	**---**	**---**	−0.003 (0.012); *p* = 0.801	−0.010 (0.016); *p* = 0.532
Eating (t2) ^f^	**---**	**---**	0.197 (0.206); *p* = 0.341	0.197 (0.301); *p* = 0.519
Drinking (t2) ^f^	**---**	**---**	−0.167 (0.221); *p* = 0.452	0.254 (0.614); *p* = 0.682
Coffein (t2) ^f^	**---**	**---**	**−0.345 (0.154); *p* = 0.027**	−0.319 (0.178); *p* = 0.082
Alcohol (t2) ^f^	**---**	**---**	−0.144 (0.157); *p* = 0.362	−0.032 (0.169); *p* = 0.848
Cigarettes (t2) ^f^	**---**	**---**	−0.166 (0.190); *p* = 0.385	0.167 (0.262); *p* = 0.526
Physical activity (t2) ^f^	**---**	**---**	0.075 (0.117); *p* = 0.522	0.025 (0.137); *p* = 0.857
Brushing teeth (t2) ^f^	**---**	**---**	0.144 (0.119); *p* = 0.230	0.042 (0.131); *p* = 0.751
Medication (t2) ^f^	**---**	**---**	−0.103 (0.153); *p* = 0.502	0.116 (0.197); *p* = 0.560
**Random effects (SD)**				
Intercept	0.76	0.585	0.636	0.691
Residual	0.706	0.774	0.545	0.439
**(B) Emotional Pain Intensity**				
**Effects**	**Stress(t2) → Emotional Pain (t2)**	**Stress(t1) → Emotional Pain (t2)**	**Emotional Pain (t2) → Oxytocin (t2)**	**Emotional Pain (t2) × Stress (t2) → Oxytocin (t2)**
**Fixed Effects**				
Intercept	**1.625 (0.360); *p* < 0.001**	**1.378 (0.308); *p* < 0.001**	**5.235 (0.318); *p* < 0.001**	**5.244 (0.583); *p* < 0.001**
Stress (t2) ^a^	**0.014(0.004); *p* < 0.001**	**---**	**---**	−0.002 (0.003); *p* = 0.474
Emotional pain ^b^	**---**	**---**	**−0.120 (0.048); *p* = 0.014**	−0.151 (0.077); *p* = 0.058
Stress (t2) x Emotional pain (t2)	**---**	**---**	**---**	−0.004 (0.003); *p* = 0.164
**Between-person**				
Stress (t1) ^a^	**---**	0.009 (0.005); *p* = 0.066	**---**	**---**
Stress (t2) ^a^	**0.026 (0.005); *p* < 0.001**	**---**	**---**	0.001 (0.004); *p* = 0.834
Emotional pain (t2) ^b^	**---**	**---**	−0.022 (0.060); *p* = 0.715	−0.065 (0.094); *p* = 0.491
Stress (t2) x Emotional pain (t2)	**---**	**---**	**---**	0.002 (0.003); *p* = 0.579
**Covariates**				
Age (t2)	**0.014 (0.007); *p* = 0.042**	0.005 (0.007); *p* = 0.426	−0.009 (0.004); *p* = 0.050	**−0.017 (0.006); *p* = 0.002**
Sex ^c^	0.092 (0.197); *p* = 0.413	0.186 (0.198); *p* = 0.352	0.018 (0.123); *p* = 0.882	−0.029 (0.150); *p* = 0.849
Partner (t2) ^d^	−0.315 (0.239); *p* = 0.641	0.034 (0.196); *p* = 0.864	0.210 (0.123); *p* = 0.091	**0.342 (0.155); *p* = 0.033**
Time point (t2) ^e^	−0.034 (0.022); *p* = 0.129	0.016 (0.016); *p* = 0.303	−0.017 (0.011); *p* = 0.120	−0.004 (0.015); *p* = 0.780
Body Mass Index (t2)	**---**	**---**	−0.014 (0.011); *p* = 0.214	−0.021 (0.014); *p* = 0.140
Eating (t2) ^f^	**---**	**---**	**0.462 (0.167); *p* = 0.006**	0.250 (0.205); *p* = 0.228
Drinking (t2) ^f^	**---**	**---**	**−0.467 (0.189); *p* = 0.015**	−0.053(0.417); *p* = 0.900
Coffein (t2) ^f^	**---**	**---**	−0.208 (0.127); *p* = 0.105	−0.179 (0.137); *p* = 0.200
Alcohol (t2) ^f^	**---**	**---**	−0.158 (0.124); *p* = 0.204	−0.131 (0.142); *p* = 0.362
Cigarettes (t2) ^f^	**---**	**---**	−0.019 (0.146); *p* = 0.898	0.266 (0.189); *p* = 0.166
Physical activity (t2) ^f^	**---**	**---**	0.059 (0.103); *p* = 0.566	0.054 (0.118); *p* = 0.651
Brushing teeth (t2) ^f^	**---**	**---**	0.173 (0.100); *p* = 0.085	0.127 (0.113); *p* = 0.268
Medication (t2) ^f^	**---**	**---**	−0.2093 (0.138); *p* = 0.132	**0.412 (0.161); *p* = 0.014**
**Random effects (SD)**				
Intercept	0.683	0.658	0.504	0.6
Residual	0.859	0.919	0.522	0.454

Note. Table depicts unstandardized coefficients (standard errors in parentheses) and p values of random intercept models on associations with (A) physical pain and (B) emotional pain. Number of observations = 143–304. Number of participants = 100–139. ^a^ momentary self-reports (using a visual analogue scale from 0 (relaxed) to 100 (stressed) and ^b^ a Likert scale ranging from 1 (little pain) to 6 (extreme pain); ^c^ 0 = male, 1 = female; ^d^ 0 = single, 1= in a relationship; ^e^ time-point = 1–12; ^f^ 0 = no, 1 = yes. Significant results are highlighted in bold.

## Data Availability

Data presented in this manuscript will be provided from the corresponding author on reasonable request.

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
