# Peer review of "Stress during the COVID-19 Pandemic Moderates Pain Perception and Momentary Oxytocin Levels"

_jcm, 2023, doi:10.3390/jcm12062333_

Round 1

Reviewer 1 Report

Thank you for the opportunity to evaluate the manuscript. The article is well written and the introduction reflects well the importance of the study, with updated references, demonstrating the current state of the art of the problem studied. In addition, the methodology presents a very robust and detailed structure. Thus, I am in favor of publishing the study with a few clarifications.

In the abstract, I believe it is important to report the quantitative data or mention the results found. Also, when talking about the conclusion of the study in the abstract, I believe it is important to make clear the context of the pandemic, which is one of the key points of the study.

Regarding the method, were there any inclusion criteria about years of education of the subjects? Speaking German was the criterion presented. However, years of schooling could interfere with understanding what was being asked for in the form. Was there any previous training for the application of the forms?

Regarding the form applied, was it a validated instrument? Was it previously sent to researchers to understand if the questions were clear?

How did you ensure that the saliva was collected at the proposed times?

Some important variables were not reported. For example: did the patient have depression, anxiety, any other neurological disorder or was he or she taking any medication that could influence oxytocin or the variables analyzed?

Author Response

Thank you for the opportunity to evaluate the manuscript. The article is well written and the introduction reflects well the importance of the study, with updated references, demonstrating the current state of the art of the problem studied. In addition, the methodology presents a very robust and detailed structure. Thus, I am in favor of publishing the study with a few clarifications.

Author response:

We want to thank the reviewer for this positive feedback.

In the abstract, I believe it is important to report the quantitative data or mention the results found. Also, when talking about the conclusion of the study in the abstract, I believe it is important to make clear the context of the pandemic, which is one of the key points of the study.

Author response:

We appreciate the reviewer’s comment and agree that the inclusion of statistics and clarification of the context improves the abstract. Therefore, we added more information on the results to the abstract.

Regarding the method, were there any inclusion criteria about years of education of the subjects? Speaking German was the criterion presented. However, years of schooling could interfere with understanding what was being asked for in the form. Was there any previous training for the application of the forms?

Regarding the form applied, was it a validated instrument? Was it previously sent to researchers to understand if the questions were clear?

Author response:

We thank the reviewer for pointing out this interesting question. Indeed, we didn’t have many inclusion criteria. A priori we aimed to reach the general population. Most participants in our sample obtained high school (25.7%) professional training (15.7%), or a college degree (47.9%). We used concise language in our daily assessments and additionally walked every participant individually through the assessment protocol. Before launching the study, we sent the items to several researchers and friends to check their comprehensibility. However, the items we used are not validated and we added this information in the limitation section of the discussion:

“However, these items might not comprehensively reflect the individual’s experience of stress and pain during times of pandemic, as standard questionnaires would.”

How did you ensure that the saliva was collected at the proposed times?

Author response:

This is an important issue in ecological momentary assessment studies as compared to saliva collection during laboratory visits. We now clarified the monitoring of the saliva collection in more detail in our manuscript: “Participants received standardized instructions on the saliva collection devices, the use of their smartphone to collect momentary ratings, and proper storage of the samples. The experimenters provided individual instructions via e-mail and by phone, constantly monitored the daily assessments, and were available to be contacted for questions.” Our constant monitoring included the following: the homepage we used for the assessment of psychological variables (sosci survey) was programmed to send us an alert email every time the participant did not start the smartphone assessment within 5 minutes. This way we could reach out to the participant to check whether there were any technical problems and to remind the participant to complete the assessment including saliva collection.

Some important variables were not reported. For example: did the patient have depression, anxiety, any other neurological disorder or was he or she taking any medication that could influence oxytocin or the variables analyzed?

Author response:

We thank the reviewer for pointing out that we missed mentioning this important information. We now included this information regarding the proportion of diagnoses in our sample:

“Of the whole sample 9.6% indicated that they have been diagnosed with depression, 3.1% with multiple diagnoses, and 1.9% with posttraumatic stress disorder.” Importantly, we controlled for the use of medication in our statistical analyses (please see table 2).

Reviewer 2 Report

Dear authors, 

Thank you for the opportunity to review this well written article. I have just one queries :

1) Why was a 2-day time frame selected for the EMA ?

Author Response

Dear authors, 

Thank you for the opportunity to review this well written article. I have just one queries :

  • Why was a 2-day time frame selected for the EMA ?

Author response:

We thank the reviewer for this positive feedback. This is indeed an important question. Before launching the EMA study, our study team thoroughly discussed the timeframe we should use to assess the EMA. We understand that 2 days might not be as representative as assessing the variables of interest, for example for one week or even longer. However, we aimed to address our research question in a broad sample and hoped to collect data from the general population. We feared that we would cause a decrease in compliance and participation of a particular group of individuals (for example elderly or more burdened participants) if we increased the collection timeframe. Above this, we figured a two-days EMA assessment would still offer the possibility to calculate correspondence of the momentary measures between days to calculate reliability scores in our outcomes.

Reviewer 3 Report

The author investigated short and long-term associations between self-reported stress levels and the intensity of physical and emotional pain in the general population during times of the Covid-19 pandemic. Additionally, The author analyzed whether salivary oxytocin levels were associated with the intensity of pain and whether this association was moderated by stress level. However, some English abbreviations in the manuscript, such as EMA, should have their full names when they first appear.

Author Response

The author investigated short and long-term associations between self-reported stress levels and the intensity of physical and emotional pain in the general population during times of the Covid-19 pandemic. Additionally, The author analyzed whether salivary oxytocin levels were associated with the intensity of pain and whether this association was moderated by stress level. However, some English abbreviations in the manuscript, such as EMA, should have their full names when they first appear.

Author response:

We thank the reviewer for this positive feedback and appreciate the comment regarding the abbreviations. We now checked the manuscript and introduced the abbreviations when it was used for the first time.